# Carbon Nanotube Coated Fibrous Tubes for Highly Stretchable Strain Sensors Having High Linearity

**DOI:** 10.3390/nano12142458

**Published:** 2022-07-18

**Authors:** Chenchen Li, Bangze Zhou, Yanfen Zhou, Jianwei Ma, Fenglei Zhou, Shaojuan Chen, Stephen Jerrams, Liang Jiang

**Affiliations:** 1College of Textiles and Clothing, Qingdao University, Qingdao 266071, China; chenchenli3@outlook.com (C.L.); bangzezhou@outlook.com (B.Z.); mjwfz@qdu.edu.cn (J.M.); fenglei.zhou@ucl.ac.uk (F.Z.); qdchshj@qdu.edu.cn (S.C.); 2Department of Medical Physics and Biomedical Engineering, University College London, London WC1V 6LJ, UK; 3Focas Research Institute, Technological University Dublin (TUD), City Campus, Kevin St, D08 NF82 Dublin, Ireland; stephen.jerrams@tudublin.ie

**Keywords:** fibrous tubes, strain sensor, working range, linearity

## Abstract

Strain sensors are currently limited by an inability to operate over large deformations or to exhibit linear responses to strain. Producing strain sensors meeting these criteria remains a particularly difficult challenge. In this work, the fabrication of a highly flexible strain sensor based on electrospun thermoplastic polyurethane (TPU) fibrous tubes comprising wavy and oriented fibers coated with carboxylated multiwall carbon nanotubes (CNTs) is described. By combining spraying and ultrasonic-assisted deposition, the number of CNTs deposited on the electrospun TPU fibrous tube could reach 12 wt%, which can potentially lead to the formation of an excellent conductive network with high conductivity of 0.01 S/cm. The as-prepared strain sensors exhibited a wide strain sensing range of 0–760% and importantly high linearity over the whole sensing range while maintaining high sensitivity with a GF of 57. Moreover, the strain sensors were capable of detecting a low strain (2%) and achieved a fast response time whilst retaining a high level of durability. The TPU/CNTs fibrous tube-based strain sensors were found capable of accurately monitoring both large and small human body motions. Additionally, the strain sensors exhibited rapid response time, (e.g., 45 ms) combined with reliable long-term stability and durability when subjected to 60 min of water washing. The strain sensors developed in this research had the ability to detect large and subtle human motions, (e.g., bending of the finger, wrist, and knee, and swallowing). Consequently, this work provides an effective method for designing and manufacturing high-performance fiber-based wearable strain sensors, which offer wide strain sensing ranges and high linearity over broad working strain ranges.

## 1. Introduction

Strain sensors, capable of transforming physical deformations into measurable electrical signals, provide the potential for important applications in a wide variety of components and disciplines including use with human skin, soft robotics, and solar energy [1,2,3]. Both conventional metallic and non-metallic semiconductor strain sensors cannot be used in the design of soft electronic devices because of their intrinsic rigid characteristics and limited working ranges [4,5,6]. Flexible strain sensors have been developed to overcome the limitations of conventional strain sensors [7]. Conductive polymer composites (CPCs) consist of flexible polymer matrices, (e.g., rubber [8], thermoplastic polyurethane [9], polydimethylsiloxane [10]) and conductive fillers, (e.g., carbon nanotubes (CNTs) [11], graphene [12] and silver nanowires/nanoparticles [13]) have been widely used to fabricate flexible strain sensors. Considering the intrinsic properties of carbon-based materials such as graphene and CNTs, they can be employed not only as fillers but also as separate materials. These materials are endowed with exceptional flexibility. Zhang et al. [14] achieved a superlubric state (a friction coefficient of nearly 0.001) on suspended graphene by applying a tensile strain of up to 0.60%. Mescola et al. [15] reported on the nanomechanics of graphene conformed on different textured silicon, in which the strain of graphene is caused by the underlying corrugated substrate below.

CPCs can be fabricated using a range of methods such as dispersing conductive fillers into flexible polymer matrices [16], depositing [17], transferring [18], or printing a conductive layer on prefabricated flexible polymer substrates [2]. However, the CPCs-based strain sensors are subject to low sensitivity and narrow working ranges because their conductive paths are often fragile and thus likely to be broken during stretching [19,20,21]. To solve these drawbacks, research has been conducted to achieve a wide working range by constructing various material configurations such as serpentine structures [22], wrinkled structures [23], and crack structures [24]. However, serpentine structures and wrinkled structures have only tolerated global but not local stretching since their behavior is consistent with spring displacement theory [25]. Accordingly, local large strains could lead to damage to the conductive paths within the devices. Crack structures, which can be generated by using brittle crack master molds or applying stress to thin films, employed microcrack theories to increase the stretchability of CPCs. Nevertheless, the production of cracks is a very complex process and working ranges in these CPCs-based strain sensors are still low [25].

Electrospinning is a simple and efficient method of generating multifunctional nanofibers [26]. Due to their large specific surface areas, fine diameters, and commendable mechanical flexibility, electrospun nanofibers have been employed to fabricate strain sensors by incorporating with conductive fillers [27,28,29]. However, previously reported strain sensors based on electrospun fibers were generally only capable of sensing small strains, since the randomly oriented nanofibers in those sensors had low stretchability. Modifying the structure of electrospun nanofibers seemed to be a good way to improve the permanence of as-obtained electrospun sensors. For instance, the previously reported TPU/CNTs fibrous strain sensors that were fabricated from electrospun TPU fibers with aligned wave-like structures exhibited a wide strain sensing range of 0–900% [30]. However, their full sensing range comprised three linear regions with the highest gauge factor of 20 occurring in the 600–900% strain range, which was still pretty low. Therefore, the design and development of electrospun TPU-based stain sensors with a wide strain sensing range, high linearity over this strain range, and high gauge factor still remains a huge challenge.

To address the above-mentioned issues, CNTs-coated TPU fibrous tubes consisting of oriented wave-like structured fibers were fabricated by a combination of electrospinning, spraying, and ultrasonic-assisted deposition in the present study. CNTs were selected as the conductive components for their one-dimensional structure, excellent conductivity, good mechanical properties, and high stability [11]. Consequently, prominent CNTs conductive networks were developed, endowing the fibrous tubes with outstanding electromechanical properties. The strain sensing performance of the TPU/CNTs fibrous tube-based sensor under one-shot stretching and repeated stretching-releasing deformations was first investigated. Thereafter, the TPU/CNTs fibrous tube-based sensor was applied to monitor human body motions such as swallowing and bending of the finger, wrist, and knee. Finally, two prototype devices utilizing the TPU/CNTs strain sensor, Bluetooth component, and battery were assembled and implanted in a wrist band and kneecap support for the detection of wrist and knee motions, respectively.

## 2. Materials and Methods

### 2.1. Materials

A commercially available TPU T3170, offering an elongation at break of 870%, was purchased from Shandong INOV New Materials Co., Ltd., Zibo, China. CNTs, with diameters of 3–15 nm, lengths of 15–30 μm, and carbon purity of 97%, were purchased from Shenzhen Turing Evolution Technology Co., Ltd., Shenzhen, China. Sulfuric acid (H_2_SO_4_), nitric acid concentrate (HNO_3_), tetrahydrofuran (THF), sodium dodecyl sulfonate (SDBS), and N,N-dimethylformamide (DMF) were provided by Sinopharm Chemical Reagent Co., Ltd., Shanghai, China.

### 2.2. Preparation of TPU/CNTs Fibrous Tubes

Carboxylated CNTs were prepared by ultrasonicating the CNTs in H_2_SO_4_/HNO_3_ (3:1 (*v*/*v*)) mixed acid with a concentration of 10 g/L for 12 h and then stirring for 8 h at 65 °C. Subsequently, the mixture was centrifuged at 6000 rev/min for 20 min before filtering through 0.22 μm filter paper. Finally, the carboxylated CNTs were washed to neutral with deionized water and then dried in a vacuum oven at 40 °C. For convenience, the carboxylated CNTs were referred to as CNTs. The CNTs aqueous suspensions with concentrations of 0.5 wt%, 1.0 wt%, 1.5 wt%, and 2.0 wt% were prepared through ultrasonic dispersion for 30 min with SDBS (1 wt%) to be employed as a surfactant.

To fabricate the TPU fibrous tubes, an 18 wt% TPU spinning solution containing a mixed solvent of THF and DMF at a ratio of 3:1 was firstly prepared, and then electrospinning was carried out by using a rotating iron wire (with a diameter of 1 mm) as the fiber collector. During the electrospinning, the TPU solution was extruded from a 22 G nozzle through a micro-extruding pump under specific conditions: an extrusion rate of 10 mm/h, a rotational speed of 250 rev/min, a distance of 15 cm between the needle and the receiver, a high voltage of 15 kV, and a spinning time of 5 min. When the spinning was completed, the TPU fibrous tube was removed from the wire, stretched to a ratio of 1.2 (20% strain). Afterward, the pre-prepared CNTs suspensions with different concentrations were sprayed onto the stretched fibrous tubes. Next, the fibrous tubes containing CNTs were ultrasonicated in the CNTs suspension with the same concentration as that for spraying for 1 h. The combination of spraying and ultrasonic assisted coating was intended to deposit more CNTs and prevent the shedding of CNTs. Figure 1a illustrates the procedure for fabricating the TPU/CNTs fibrous tubes. Using this process, CNTs coated TPU fibrous tubes with oriented and wavy structured fibers were obtained, as shown in Figure 1b–d.

### 2.3. Characterization

The surface and cross-section morphology of TPU/CNTs fibrous tubes was observed by using a scanning electron microscope (SEM, TESCAN VEGA 3, Brno-Kohoutovice, Czech Republic). Prior to the characterization, a thin layer of gold was coated on the samples and then SEM images with different magnifications were captured under an acceleration voltage of 10 kV.

The surface chemical composition of the carboxylated CNTs was measured through an X-ray photoelectron spectroscopy (XPS, Axis Supra+, Kyoto, Japan). The Al Kα X-ray source was operated at 15 kV and 10 mA, and the C 1s peak was shifted to 284.8 eV for energy calibration.

The interface interactions of TPU/CNTs were studied using a Fourier transform infrared spectroscopy (FTIR, Nicolet iS10, Waltham, MA, USA). The spectrum was collected in the wavenumber range of 4000–500 cm^−1^ at scanning times of 32 and a resolution of 4 cm^−1^.

Thermogravimetric tests were performed in a nitrogen atmosphere using a DSC/TG synchronous thermal analyzer (STA449F3 Jupiter, Bavaria, Germany). Approximately 5 mg samples were heated from ambient temperature to 800 °C at a heating rate of 20 °C·min^−1^.

The stress–strain curves were obtained from tests using a universal tester (Instron 5965, Norwood, MA, USA) at a stretching rate of 100 mm/min. The test samples were TPU/CNTs fibrous tubes with lengths of 40 mm and the gauge distance was 20 mm.

The real-time resistance changes of the TPU/CNTs fibrous tube-based sensors under stretching were recorded using a digital multimeter (Keysight B2901A, Keysight Technology, Santa Rosa, CA, USA). TPU/CNTs fibrous tubes of 40 mm length were wrapped with copper tapes at each end, and the tensile deformation was induced by a stepper motor. Furthermore, the real-time resistance changes of the TPU/CNTs fibrous tube-based sensors in response to swallowing and finger, wrist, elbow, and knee bending were also measured with the digital multimeter.

## 3. Results and Discussion

### 3.1. Microstructural Characterization

Figure 2 presents the surface morphologies of the resultant TPU/CNTs tubes. As depicted, the TPU fibers produced by electrospinning showed a tendency of orienting axially along the tubes, primarily because the nozzle moved parallel to the wire collector during the electrospinning process. Furthermore, most fibers achieved a curved morphology, which could have two explanations. The fibers were pre-stretched during the spraying process with the CNTs suspension and then the stretching was released after spraying. The purpose of this was to allow more CNTs to attach to the fibers and thus form an effective conductive network. Secondly, when the fibrous tubes were immersed in the CNTs suspension, they partially softened and curved under the action of high-energy microjets and shock waves during ultrasonication. Figure 2 reveals that when CNTs were coated onto the TPU fibrous tubes, a tufted surface formed. CNTs were also observed in the interfiber pores since the collapse of air bubbles during ultrasonication forced CNTs into the internal fiber pores [31,32]. With the increase in the concentration of the CNTs suspension, the number of CNTs on the TPU fibrous tubes increased, and CNTs agglomerates became apparent.

### 3.2. Chemical Structure Characterization and Physical Characterization

To facilitate the formation of a strong interaction between the CNTs and TPU, CNTs were carboxylated before use [33]. From the XPS test (Figure 3a,b), it can be seen that CNTs had only a singular main peak of C 1s, and carboxylated CNTs had two main peaks of C 1s and O 1s, with binding energies of 284.8 and 532 eV, respectively. As depicted in Figure 3b, the C 1s core-level spectrum of the CNTs could be curve fitted with three peak components having binding energies of 284 eV (sp^2^ C), 284.8 eV (C–C), and 291.5 eV (π–π* peak). An additional peak component at 289 eV for O–C=O appeared in the C 1s core-level spectrum of carboxylated CNTs. This confirms the existence of the O–C=O group in carboxylated CNTs.

As revealed by the FTIR results (Figure 3c), a few peaks for TPU fibers shifted after the deposition of the carboxylated CNTs. For instance, the peak of the C–H band shifted from 2955 cm^−1^ to 2950 cm^−1^, and the vibration at around 1726 cm^−1^ for C=O shifted to 1725 cm^−1^, which suggested the formation of hydrogen bonds between TPU and carboxylated CNTs [30].

Thermal stability has been found to play a vital role in the applications of composite materials [32]. To clarify the thermal behavior and the specific amount of CNTs deposited onto the TPU fibrous tubes, TG analysis was conducted, as presented in Figure 3d. The figure shows that the decomposition processes for TPU and TPU/CNTs were similar. The thermal degradation of the TPU began with the breakage of hydrogen bonds generated by N–H and C=O groups, followed by the decomposition of hard and soft blocks [34]. Compared with TPU, TPU/CNTs were found to have higher initial decomposition temperatures since hydrogen bonds between the carboxyl-modified CNTs and the TPU were formed, thus enhancing the thermal stability of the TPU. However, impacted by the high thermal conductivity of the CNTs, TPU/CNTs exhibited lower final decomposition temperatures than those of pure TPU. The actual contents of the CNTs in TPU/0.5 wt% CNTs, TPU/1.0 wt% CNTs, TPU/1.5 wt% CNTs, and TPU/2.0 wt% CNTs fibrous tubes were measured at 8 wt%, 10 wt%, 12 wt%, and 13 wt%, respectively.

The mechanical properties exhibited by the TPU/CNTs fibrous tubes were investigated to determine their feasibility to act as components for wearable strain sensors. As revealed by the stress–strain curves (Figure 3e), all TPU/CNTs fibrous tubes achieved tensile strengths of higher than 10 MPa and elongations at a break of higher than 600%. It is noteworthy that the increase in stress for all TPU/CNTs fibrous tubes was virtually linear as the strain increased. This was possibly because the wavy and aligned structure endowed the tubes with uniform mechanical properties. The ultimate tensile strength (UTS) decreased with the increase in CNTs content since the strong van der Waals force between the CNTs limited the movement of the TPU molecular chains [35]. Both the TPU/1.5 wt% CNTs and the TPU/2.0 wt% CNTs had elongations at break close to 800%. Each fibrous tube exhibited a UTS of approximately 12 MPa to 13 MPa. These values showed that the fibrous tube-based strain sensors could change electrical response signals over wide working ranges.

### 3.3. Electrical Property

Figure 4a depicts the dependency of relative electrical resistance change (*ΔR/R*_0_**, where *R*_0_** denotes the original resistance and *R* represents the real-time resistance during stretching) on strain for TPU/CNTs fibrous tubes. As shown in the figure, *ΔR/R*_0_** increased when the strain increased for all TPU/CNTs fibrous tubes as the applied stress increased the tunneling distance between the CNTs, resulting in the breakage of conductive pathways [36]. All the fibrous tube-based strain sensors achieved electrical responses to large strains of greater than 300%. With the increase in the concentration of the CNTs suspension in the range of 0.5 wt%–1.5 wt%, the strain sensing range of the resultant TPU/CNTs fibrous tube was expanded. In particular, the strain sensor based on TPU/CNTs fibrous tube prepared with 1.5 wt% CNTs suspension was capable of sensing strains as high as 760%. This sensing capacity is superior to most of those strain sensors reported recently (Figure 4d) [2,3,4,5,6,7,10,11,13,15,17,21,30,37,38,39,40,41,42]. The TPU/2.0 wt% CNTs conductive tubes could also sense strains up to 710%. The slightly lower sensing strain obtained by TPU/2.0 wt% CNTs fibrous tube than that of TPU/1.5 wt% CNTs was consistent with its lower elongation at break, as can be observed in Figure 3e. In addition, the gauge factor (*GF = (ΔR/R*_0_*)/ε, ε* denotes the applied strain) employed to assess the sensitivity of strain sensors [15,43,44] also achieved a high value of 57 for the TPU/1.5 wt% CNTs fibrous tube (Figure 4c), which is superior to most reported strain sensors. Though some of the strain sensors with high gauge factors reported in the literature were able to monitor both very small and large motions, the signal produced by very small motions might disrupt the desired signal.

Achieving high linearity also takes on a great significance for strain sensors in simplifying the calibration processes and increasing the accuracy of electrical signals. The coefficient of determination (*r^2^*) was calculated to characterize the linearity of TPU/CNTs fibrous tube-based strain sensors. Figure 4b depicts *r^2^* in the strain ranges from 0 to 300% and from 0 to the maximum working strain for all TPU/CNTs fibrous tube-based strain sensors. As revealed by the results, the tubes had *r^2^* values higher than 0.9 in their working strain ranges, which suggested excellent linearity. It is noteworthy that an *r^2^* of 0.9857 was achieved for the strain sensing range of 0–760% for TPU/1.5 wt% CNTs fibrous tubes-based stain sensor.

The mechanism for the large sensing strain of the TPU/CNTs fibrous tube-based sensor is illustrated diagrammatically in Figure 4(ei–eiii). When a small stretch was applied to the fibrous tubes, the undulating fibers (Figure 4ei) were straightened in the direction of the tube axis (Figure 4(eii)). With further stretching, the fibers became more aligned, and some interfiber connections were formed. Since CNTs were sprayed onto the TPU fibrous tubes in the stretched state, compact CNTs networks were developed once stretching was released. Accordingly, the CNTs conductive network was kept intact during stretching, thus allowing good electrical conductivity at large strains and ensuring a wide strain sensing range (Figure 4(eiii)).

Mechanical hysteresis significantly affects the sensing stability of elastomer-based strain sensors [43]. Accordingly, the mechanical hysteresis of TPU/CNTs fibrous tubes was studied. As depicted in Figure 5a,d, the mechanical hysteresis was minimal when the TPU/1.5 wt% CNTs were stretched to small strains (2–10%), while it was significantly larger when the sample was subjected to larger tensile strains (50–400%). Figure 5b,e illustrates the mechanical hysteresis of the respective tensile strain cycles. These were obtained by calculating the area between the loading and unloading curves of the corresponding cycles. The mechanical hysteresis of the TPU/1.5 wt% CNTs composite increased with the increase in strain. However, hysteresis tended to stabilize after the first two cycles and changed very little thereafter. This is typical for most elastomers as reported in the literature [30].

Figure 5c,f depicts the change of Δ*R/R*_0_** in 30 stretching–releasing cycles with strains ranging from 0 to diverse peak strains. The stretching (or releasing) rate was 10 mm/min. For all samples, in the respective cycle, the peak value of *ΔR/R*_0_** increased with increasing strain. In the first cycle, the initial value of Δ*R/R*_0_** was lower than the final value after unloading. This was because the CNTs conductive networks did not recover completely due to mechanical hysteresis, the interaction between the CNTs and TPU [45], and the tendency for the set to occur in elastomers as a result of repeated tensile cycles. For tensile cycles over the same strain range, the peak value of *ΔR/R*_0_** tended to decrease in the first few cycles before stabilizing. This was consistent with the continuous destruction and reconstruction of the CNTs network establishing a stable conductive path [46].

Figure 5g,h illustrates the current–voltage curves of TPU/1.5 wt% CNTs fibrous tube at different tensile strains. Under specific voltages, the resistance of the TPU/1.5 wt% CNTs fibrous tube increased with an increase in the strain. The linearity of the curves revealed excellent Ohmic characteristics and high electrical conductivity in a wide strain range from 2% to 400%. Figure 5i shows the dynamic response of the TPU/1.5 wt% CNTs strain sensor under a 2% strain. For an instantaneous strain of 2%, the material responded quickly within 45 ms. These results suggest that the TPU/1.5 wt% CNTs fibrous tube-based strain sensor had a wide sensing range, high sensitivity, excellent repeatability, and rapid response time.

### 3.4. Stability and Durability

The stability and durability of the sensors were characterized by exposing them to 10,000 stretching–releasing cycles at a constant strain range of 0–200%. The stretching (or releasing) speed was 200 mm/min. As depicted in Figure 6a, although the strain sensor indicated overshoot in the first few cycles due to the reconstruction and stabilization of the CNTs networks, it exhibited excellent stability and durability in the following cycles until the end of the test.

Washing stability takes on a critical significance for wearable strain sensors. To explore the washing fastness of TPU/1.5 wt% CNTs fibrous tubes, they were immersed in distilled water under ultrasonication for 0 min, 20 min, 40 min, and 60 min, and the electrical resistance and sensing ability were determined after drying. With an extension in the ultrasonic washing time, the resistance of the TPU/1.5 wt% CNTs fibrous tubes hardly changed (Figure 6b). As depicted in Figure 6c, after being washed for 60 min, the TPU/1.5 wt% CNTs fibrous tubes-based strain sensor was still capable of sensing strains of up to 760%. Furthermore, the change of *ΔR/R*_0_** during cyclic stretching-releasing after washing (Figure 6d) was virtually the same as that before washing. Hence, the TPU/1.5 wt% CNTs fibrous tubes exhibited excellent washing fastness.

### 3.5. Application of the TPU/1.5 wt% CNTs Strain Sensor

With a wide working range, high sensitivity, and reliable durability, the TPU/1.5 wt% CNTs fibrous tube-based strain sensor was used to detect various human motions. A consent form was read and signed by the volunteer before the tests. For large motion sensing, the strain sensors were attached to finger, wrist, elbow, and knee joints. As depicted in Figure 7a–d, when these joints underwent reciprocating bending from their normal positions, the strain sensor could give an instantaneous response through the observed resistance change. In addition, the response signals differed for different joints, which suggested that the strain sensor had the ability to discriminate between different kinds of joint motions. The strain sensor was also attached to the throat for capturing subtle physiological signals. Figure 7e clearly shows that *ΔR/R*_0_** changed with the action of swallowing, which suggested that the sensor could detect subtle human body motions. Figure 7f and Appendix A illustrate the use of TPU/1.5 wt% CNTs fibrous tube for regulating the brightness of LED lights. When the TPU/1.5 wt% CNTs fibrous tube was stretched from 0 to 600% at a velocity of 10 mm/min, the brightness of the connected LED lights dimmed. This demonstrates the tensile resistance response behavior of the TPU/1.5 wt% CNTs fibrous tube-based strain sensor.

To further demonstrate the sensing ability of TPU/CNTs fibrous tubes, a smart motion detector was assembled by integrating the TPU/1.5 wt% CNTs strain sensor, a USB screen tester, and a lithium battery. Figure 8a presents the schematic diagram of the intelligent motion detection device and Figure 8b depicts a wrist band and kneecap support implanted with the intelligent motion detection device. The current signals generated by the movement of volunteers’ wrists and knees could be transmitted to a smartphone in real time via a Bluetooth signal generated by the USB screen tester. Figure 8c,e,f,h–j and Appendix A illustrate the application of the intelligent motion detection device in monitoring wrist and knee movements. A constant current signal was received by the smartphone (Figure 8d,g) as both the wrist and the knee were kept static (Figure 8c,h). When the wrist was repeatedly flexed (Figure 8e), the current signal received by the smartphone changed repeatedly (Figure 8f). When the knee was bent repeatedly (Figure 8i), the current signal received by the smartphone changed spontaneously (Figure 8j). The motion detection results for the wrist and knee bending confirmed the excellent sensing performance and repeatability of the TPU/1.5 wt% CNTs sensor.

## 4. Conclusions

In this work, an electrically conductive TPU/CNTs fibrous tube was prepared, and its feasibility for constructing highly stretchable strain sensors was investigated. As a result of the aligned fiber orientation of the TPU fibrous tubes and the formation of effective CNTs conductive networks, the resultant TPU/CNTs fibrous tube-based strain sensor exhibited a wide strain sensing range of 0–760%, high linearity over the whole sensing range and high sensitivity with a *GF* of 57. Furthermore, the strain sensor demonstrated an ability to detect low strains (2%) and a fast response in the region of 45 ms with high durability. The TPU/CNTs fibrous tube-based strain sensor was found to be capable of accurately monitoring both large and subtle human body motions, (e.g., finger, wrist, elbow, and knee bending, and swallowing). The findings of this work provide the opportunity for developing flexible strain sensors exhibiting high stretchability, low detection limits, and fast response capability.

## Figures and Tables

**Figure 1 nanomaterials-12-02458-f001:**
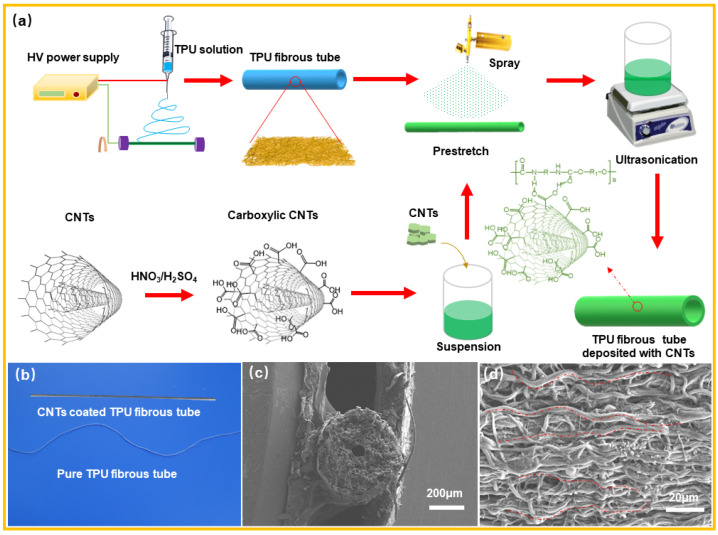
(**a**) A diagram depicting the preparation of a TPU/CNTs fibrous tube; (**b**) a depiction of a CNTs coated TPU fibrous tube and a pure TPU fibrous tube; (**c**) the surface and (**d**) the cross-section morphology of the CNTs coated TPU fibrous tube, respectively.

**Figure 2 nanomaterials-12-02458-f002:**
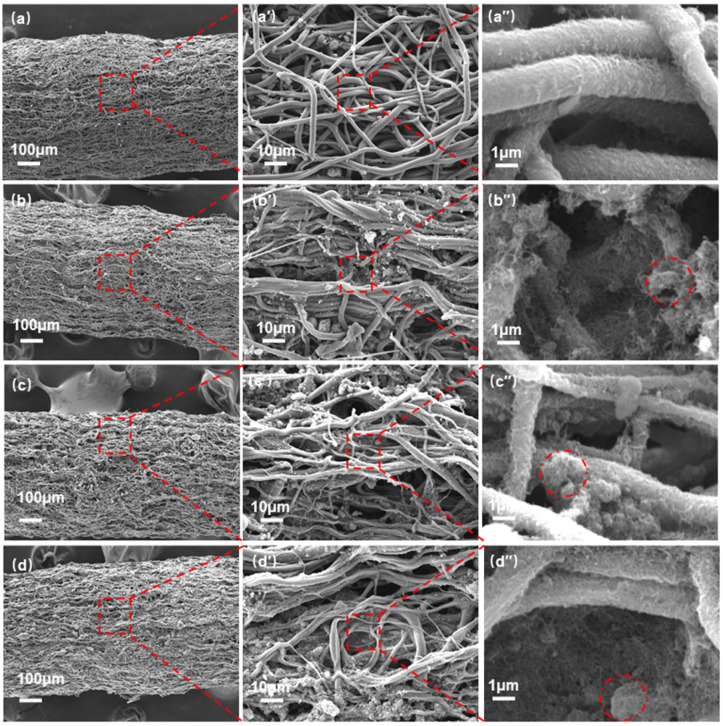
SEM images of the TPU fibrous tubes prepared with CNTs concentrations of (**a**–**a**″) 0.5 wt%, (**b**–**b**″) 1.0 wt%, (**c**–**c**″) 1.5 wt%, and (**d**–**d**″) 2.0 wt%, (the areas enclosed by the red dash lines represent CNT agglomerates).

**Figure 3 nanomaterials-12-02458-f003:**
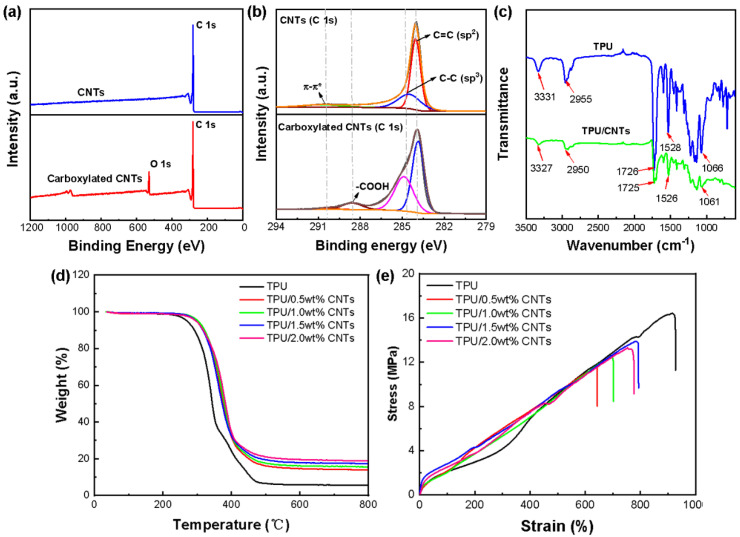
(**a**) XPS wide scan spectra of the CNTs and carboxylated CNTs; (**b**) C 1s core–level spectra of the CNTs and carboxylated CNTs; (**c**) FTIR spectra of the CNTs and carboxylated CNTs; (**d**) TG curves and (**e**) typical stress–strain curves of TPU/0.5 wt% CNTs, TPU/1.0 wt% CNTs, TPU/1.5 wt% CNTs and TPU/2.0 wt% CNTs.

**Figure 4 nanomaterials-12-02458-f004:**
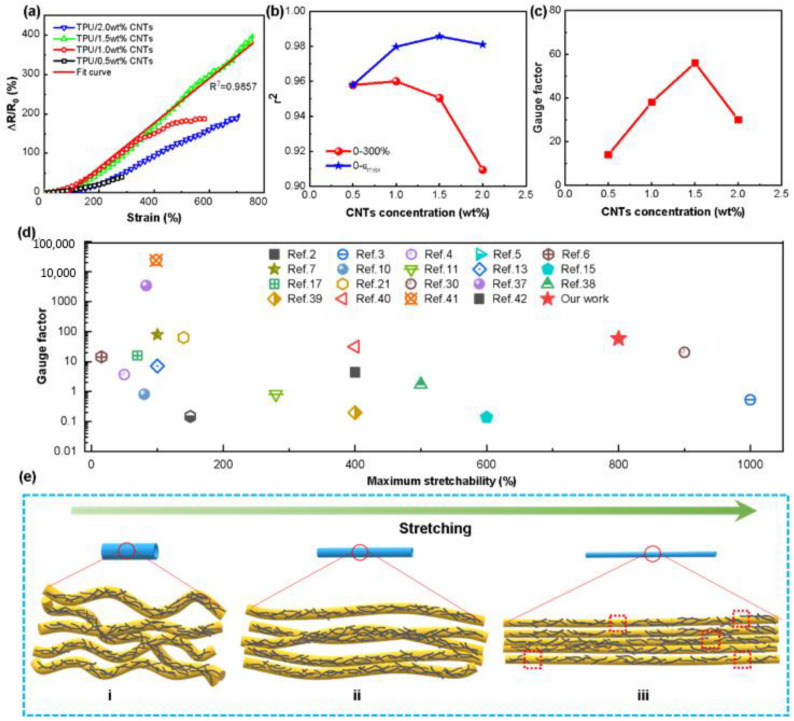
(**a**) Plots of relative electrical resistance change versus strain for TPU/0.5 wt% CNTs, TPU/1.0 wt% CNTs, TPU/1.5 wt% CNTs, TPU/2.0 wt% CNTs and the relative resistance–strain fitting curve of TPU/1.5 wt% CNTs; (**b**) the *r^2^* of TPU/CNTs tube prepared with various CNTs concentrations; (**c**) the relation between *GF* of TPU/CNTs tube and CNTs concentration; (**d**) comparison of the maximum *GF* and the maximum sensing strain reported in the literature and the research described in this text; (**e**) schematic illustration of the changes in TPU/CNTs fibers during stretching.

**Figure 5 nanomaterials-12-02458-f005:**
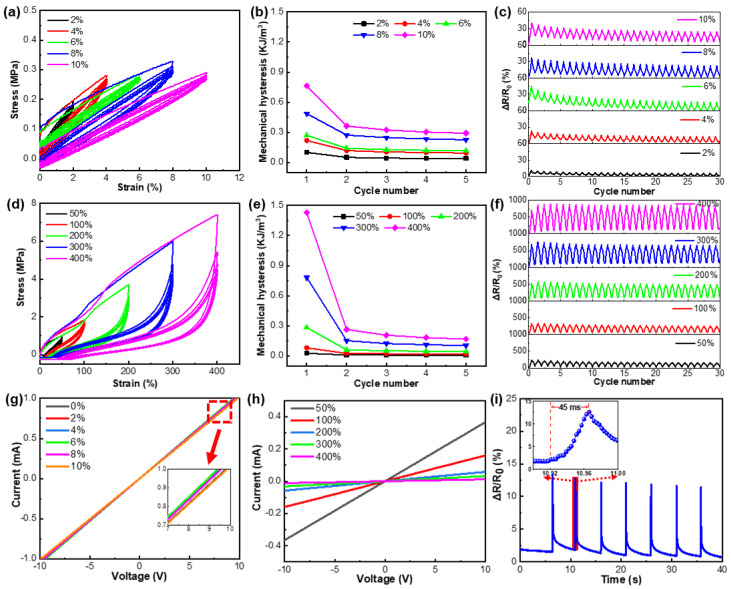
Hysteresis curves of the initial five tensile strain cycles of the TPU/1.5 wt% CNTs fibrous tube (**a**) at low strains of 2–10% and (**d**) at high strains of 50–400%; the mechanical hysteresis in five cycles for TPU/1.5 wt% CNTs at (**b**) small strains of 2–10% and (**e**) large strains of 50–400%; current–voltage curves of TPU/1.5 wt% CNTs strain sensors at strains of (**g**) 0–10% and (**h**) 50–400%; dynamic response behavior of TPU/1.5 wt% CNTs sensors during tensile strain cycles at different peak strains of (**c**) 2–10% and (**f**) 50–400%; (**i**) The response time of the TPU/1.5 wt% CNTs strain sensors, the insert shows a quick response of about 45 ms.

**Figure 6 nanomaterials-12-02458-f006:**
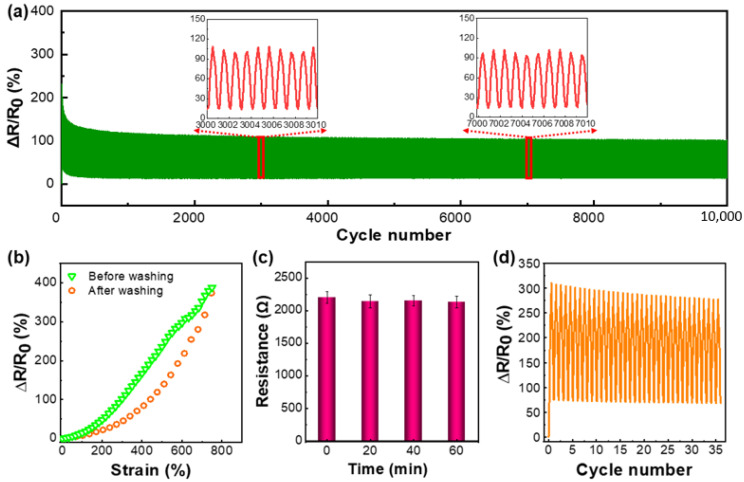
(**a**) Relative resistance change of the TPU/1.5 wt% CNTs strain sensor in 10,000 stretching-releasing cycles under a strain range of 0–200%; (**b**) relative resistance change–strain curve at a feed rate of 10 mm/min for the TPU/1.5 wt% CNTs fibrous tubes after washing; (**c**) the resistance of TPU/1.5 wt% CNTs fibrous tubes after different washing times; (**d**) the change of relative resistance of TPU/1.5 wt% CNTs fibrous tubes after washing under cyclic tensile loading where the strain range was 0–100% and the feed rate was 10 mm/min.

**Figure 7 nanomaterials-12-02458-f007:**
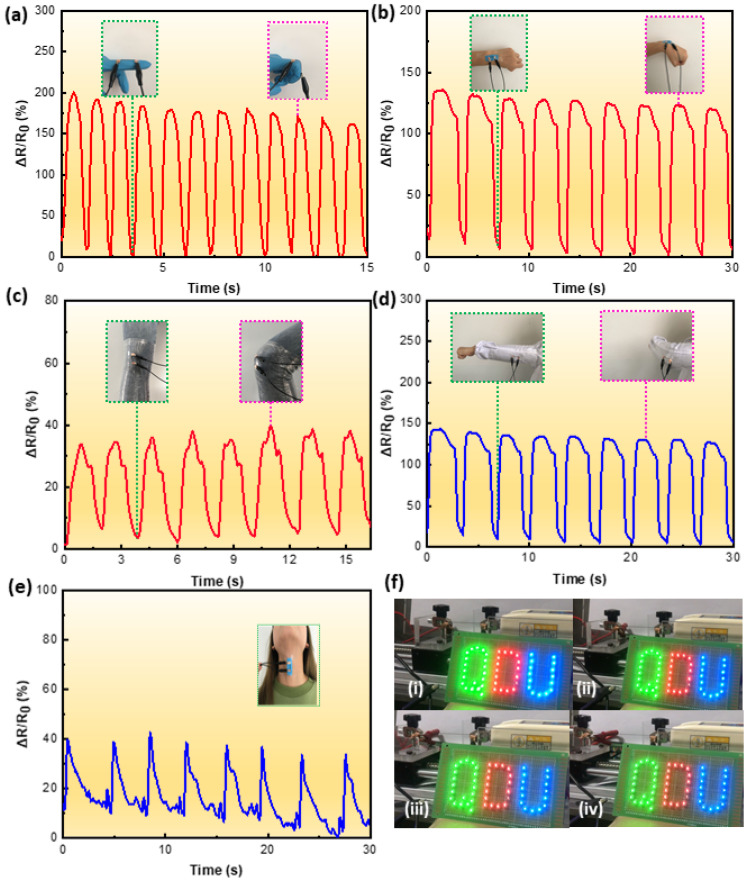
Real-time relative resistance change to detect (**a**) index finger joint bending, (**b**) wrist bending, (**c**) leg squatting and lifting, (**d**) arm bending, and (**e**) swallowing; (**f**) changes in brightness of LED with a conductive TPU/1.5 wt% CNTs fibrous tube stretched from 0 to 600%.

**Figure 8 nanomaterials-12-02458-f008:**
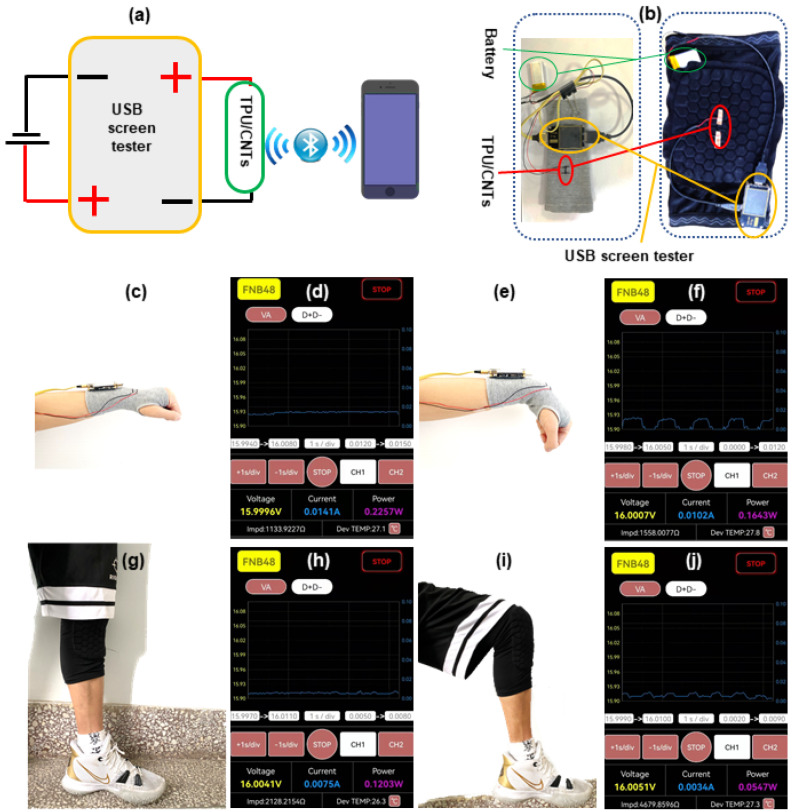
(**a**) Schematic diagram of intelligent motion detection device; (**b**) a photograph of wrist band and kneecap support implanted with the intelligent motion detection device, respectively; the photographs of wrist (**c**) resting and (**e**) bending, and (**d**,**f**) the corresponding current signal; the photographs of knee (**g**) resting and (**i**) bending, and (**h**,**j**) the corresponding current signal.

## Data Availability

The data presented in this study are available on request from the corresponding author.

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
