# Peer review of "Carbon Nanotube Coated Fibrous Tubes for Highly Stretchable Strain Sensors Having High Linearity"

_nanomaterials, 2022, doi:10.3390/nano12142458_

Round 1

Reviewer 1 Report

In the research article titled "Coating of thermoplastic polyurethane fibrous tubes with carbon nanotubes to facilitate the fabrication of highly stretchable strain sensors" C. Li et al. describe the fabrication of a flexible strain sensor based on conductive polymer composite - thermoplastic polyurethane fibers coated with carboxilated multiwall carbon nanotubes. The presented sensor has been deeply characterized from chemical, mechanical and electrical point of view. In addition, the sensor's performance has been tested under one-shot and repeated stretching-releasing deformation and also applied to human body (finger, wrist and knee) to monitor motions demonstrating its ability to detect both large and subtle human motions.

The manuscript is clear, well-written and well-organized. Figures are exhaustiveand clear. The results are interesting with a high potential of disruptive applications. However, the manuscript should be implemented in some point before considering it suitable for nanomaterials journal; in particular, the discussion section is lacking as well as the introduction should be slightly broaden. Recommendations to improve the quality of the manuscript are listed below:

- At line 40, authors introduce CPCs and their use in the field of flexible strain sensors. I do think few info about the intrinsic properties of carbon-based materials, such as graphene and CNTs – not only as fillers but also as separate materials – should be added. They are endowed with peculiar flexibility; graphene under strain, for example, has been recently investigated (please, consider the following articles: PNAS, 2020 - 24452–24456 https://doi.org/10.1073/pnas.1907947116; Small 2021, 17, 2104487 https://doi.org/10.1002/smll.202104487;) and the possibility to achieve certain levels of strain, in the presence and in the absence of an underling substrate has been proved. Such a mention would emphasize even more the powerful of composites properties giving a quantitative idea about the strain range achievable with such a kind of materials.

-A specific discussion section is missing. Authors can decide to merge it in ‘Results and Discussion’ but few comments should be added to broaden the discussion.

-At lines 153-54, authors state: Firstly, the fibers were pre-stretched during the spraying process with the CNTs suspension and then released after spraying. Could you add the reason why? There is a technical reason behind? Please, specify.

-At lines 167-68, authors state: To facilitate the formation of a strong interaction between the CNTs and TPU, CNTs were carboxylated before use. You should mention the case where CNTs not carboxilated provide poor interaction with TPU. References should be added here.

-At lines 202-3, authors state: The TPU/1.5 wt% CNTs and the TPU/2.0 wt% CNTs had elongations at break close to 700% and 800%, respectively. From panel 3e, it seems slightly lower (650%-700%).

-In the Figure 4, panel a, data of TPU/0.5 wt% CNTs are not visible in the printed version: May you try to bring them to the forefront.

Author Response

Reviewer 1#

In the research article titled "Coating of thermoplastic polyurethane fibrous tubes with carbon nanotubes to facilitate the fabrication of highly stretchable strain sensors" C. Li et al. describe the fabrication of a flexible strain sensor based on conductive polymer composite - thermoplastic polyurethane fibers coated with carboxilated multiwall carbon nanotubes. The presented sensor has been deeply characterized from chemical, mechanical and electrical point of view. In addition, the sensor's performance has been tested under one-shot and repeated stretching-releasing deformation and also applied to human body (finger, wrist and knee) to monitor motions demonstrating its ability to detect both large and subtle human motions.

The manuscript is clear, well-written and well-organized. Figures are exhaustive and clear. The results are interesting with a high potential of disruptive applications. However, the manuscript should be implemented in some point before considering it suitable for nanomaterials journal; in particular, the discussion section is lacking as well as the introduction should be slightly broaden. Recommendations to improve the quality of the manuscript are listed below:

- At line 40, authors introduce CPCs and their use in the field of flexible strain sensors. I do think few info about the intrinsic properties of carbon-based materials, such as graphene and CNTs – not only as fillers but also as separate materials – should be added. They are endowed with peculiar flexibility; graphene under strain, for example, has been recently investigated (please, consider the following articles: PNAS, 2020 - 24452–24456 https://doi.org/10.1073/pnas.1907947116; Small 2021, 17, 2104487 https://doi.org/10.1002/smll.202104487;) and the possibility to achieve certain levels of strain, in the presence and in the absence of an underling substrate has been proved. Such a mention would emphasize even more the powerful of composites properties giving a quantitative idea about the strain range achievable with such a kind of materials.

Response: The authors thank the reviewer for this kind suggestion. Some of the recommended references have been added in the revised manuscript.

-A specific discussion section is missing. Authors can decide to merge it in ‘Results and Discussion’ but few comments should be added to broaden the discussion.

Response: The authors thank the reviewer for this comment. The paper has been modified in the revised manuscript.

-At lines 153-54, authors state: Firstly, the fibers were pre-stretched during the spraying process with the CNTs suspension and then this pre-stretching was released after spraying. Could you add the reason why? There is a technical reason behind? Please, specify.

Response: The fibers were pre-stretched during the spraying process with the CNTs suspension and then the stretching was released after spraying. The purpose for this was to allow more CNTs to attach to the fibers and thus form an effective conductive network.

-At lines 167-68, authors state: To facilitate the formation of a strong interaction between the CNTs and TPU, CNTs were carboxylated before use. You should mention the case where CNTs not carboxilated provide poor interaction with TPU. References should be added here.

Response: The authors thank the reviewer for this kind suggestion. The reason for the carboxylation of CNTs has been added with one reference cited in the revised manuscript.

-At lines 202-3, authors state: The TPU/1.5 wt% CNTs and the TPU/2.0 wt% CNTs had elongations at break close to 700% and 800%, respectively. From panel 3e, it seems slightly lower (650%-700%).

Response: The authors thank the reviewer for this kind suggestion. We made a mistake to the figure when we revised the paper according to our collaborators’ suggestion. As we can know from the paragraph above the figure “The ultimate tensile strength (UTS) decreased with the increase in CNTs content since the strong van der Waals force between the CNTs limited the movement of the TPU molecular chains”, thus the curves displayed in Figure 3(e)is undoubtedly wrong. Now, it has been corrected. Besides, the sentence “The TPU/1.5 wt% CNTs and the TPU/2.0 wt% CNTs had elongations at break close to 700% and 800%, respectively.” has been changed to “Both the TPU/1.5 wt% CNTs and the TPU/2.0 wt% CNTs had elongations at break close to 800%.”

-In the Figure 4, panel a, data of TPU/0.5 wt% CNTs are not visible in the printed version: May you try to bring them to the forefront.

Response: The authors thank the reviewer for this kind suggestion. It has been modified in the revised manuscript.

Reviewer 2 Report

Authors have highlighted the emerging and core issue, but still there are major issues to be fixed.

Reviews to Authors

·       Title must be simple, clearer and nicer.

·       Spell out each acronym the first time used in the body of the paper. Spell out acronyms in the Abstract by extending it.

·       The abstract can be rewritten to be more meaningful. The authors should add more details about their final results in the abstract. Abstract should clarify what is exactly proposed (the technical contribution) and how the proposed approach is validated.

·       What is the motivation of the proposed work?

·       Introduction needs to explain the main contributions of the work clearer.

·       The novelty of this paper is not clear. The difference between present work and previous Works should be highlighted.

·       Authors must explain in detail the introduction section.

·       Authors must develop the framework/architecture of the proposed methods

·       There is need of flowchart and pseudocode of the proposed techniques

·       Proposed methods should be compared with the state-of-the-art existing techniques

·       Research gaps, objectives of the proposed work should be clearly justified.

To improve the Related Work and Introduction sections authors are highly recommended to consider these high-quality research works <A Compact High-Gain Coplanar Waveguide-Fed Antenna for Military RADAR Applications >

·       English must be revised throughout the manuscript.

·       Limitations and Highlights of the proposed methods must be addressed properly

·       Experimental results are not convincing, so authors must give more results to justify their proposal.

Finally, paper needs major improvements

Author Response

Reviewer 2#

Title must be simple, clearer and nicer.

Response: The authors thank the reviewer for the useful suggestion. Title changed to: Carbon nanotube coated fibrous tubes for highly stretchable strain sensors having high linearity.

Spell out each acronym the first time used in the body of the paper. Spell out acronyms in the Abstract by extending it.

Response: The full term before using acronyms has been addressed in the Abstract.

The abstract can be rewritten to be more meaningful. The authors should add more details about their final results in the abstract. Abstract should clarify what is exactly proposed (the technical contribution) and how the proposed approach is validated.

Response: The authors have revised the Abstract and made appropriate changes.

What is the motivation of the proposed work?

Response: The authors thank the reviewer for pointing this out. It has been included in Abstract.

Introduction needs to explain the main contributions of the work clearer.

Response: The research described the development of a conductive polymer composite based strain sensor with a wide strain sensing range and high linearity over this strain range. The contributions of the work have been included in the Introduction.

The novelty of this paper is not clear. The difference between present work and previous Works should be highlighted.

Response: The novelty of this work is listed as following:

   Fibrous tube comprising oriented wave-like structured polyurethane fibers were fabricated.

The fibrous tube based strain sensor showed an ultra-wide working range of 0-760% with a gauge factor of 57.

The strain sensor had fast response time of 45 milliseconds with reliable long-term stability and washing durability.

This has been addressed the Abstract.

Authors must explain in detail the introduction section.

Response: The novelty has been included in the last paragraph of the Introduction.

Authors must develop the framework/architecture of the proposed methods

Response: The frame-work of the proposed method has been included in Figure 1.

There is need of flowchart and pseudocode of the proposed techniques

Response: The flowchart and pseudocode of the proposed techniques has been included in Figure 1.

Proposed methods should be compared with the state-of-the-art existing techniques

Response: The proposed methods were compared with the state-of-the-art existing techniques as shown in Figure 4.

Research gaps, objectives of the proposed work should be clearly justified.

To improve the Related Work and Introduction sections authors are highly recommended to consider these high-quality research works <A Compact High-Gain Coplanar Waveguide-Fed Antenna for Military RADAR Applications >

Response: This has been cited in the Introduction.

English must be revised throughout the manuscript.

Response: The language has been polished by a native English speaker.

Limitations and Highlights of the proposed methods must be addressed properly

Response: They have been highlighted in Abstract and Conclusion.

Experimental results are not convincing, so authors must give more results to justify their proposal.

Response: All experimental results were obtained based on rigorous and appropriate testing by using at least 5 samples. This intensity of testing was applied to ensure the credibility of the results.

Finally, paper needs major improvements

Response: The improvements have been made to the manuscript.

Reviewer 3 Report

The interest of this research is focused on deformation sensors based on thermoplastic polyurethanes (TPU) electrospun with multi-walled carbon nanotubes, preliminarily carboxylated.

Specifically, authors explored the use of CNTs coated TPU fibrous tubes consisting of oriented wave-like structured fibres and fabricated by a combination of electrospinning, spraying and ultrasonic assisted deposition to monitor human body motions such as swallowing and bending of the finger, wrist and knee.

New strain sensors showed, among others, a sensing capacity superior to that detected for similar systems, the ability to monitor even very small movements and high linearity over the whole sensing range.

The manuscript is well organized and the results, obtained with appropriate techniques, have been clearly reported even with the support of an adequate number of figures. Discussion of the data is appropriate.

As far as I'm concerned, there are no relevant comments and / or suggestions to report and, therefore, given the repercussions that new knowledge achieved on this topic can have on the design and development of high performance wearable strain sensors, it is recommended that this paper be published in the present form.

Author Response

Reviewer 3#

The interest of this research is focused on deformation sensors based on thermoplastic polyurethanes (TPU) electrospun with multi-walled carbon nanotubes, preliminarily carboxylated.

Specifically, authors explored the use of CNTs coated TPU fibrous tubes consisting of oriented wave-like structured fibres and fabricated by a combination of electrospinning, spraying and ultrasonic assisted deposition to monitor human body motions such as swallowing and bending of the finger, wrist and knee.

New strain sensors showed, among others, a sensing capacity superior to that detected for similar systems, the ability to monitor even very small movements and high linearity over the whole sensing range.

The manuscript is well organized and the results, obtained with appropriate techniques, have been clearly reported even with the support of an adequate number of figures. Discussion of the data is appropriate.

As far as I'm concerned, there are no relevant comments and / or suggestions to report and, therefore, given the repercussions that new knowledge achieved on this topic can have on the design and development of high performance wearable strain sensors, it is recommended that this paper be published in the present form.

Response: The authors thank the reviewer for taking the time to review this work.

Round 2

Reviewer 1 Report

Authors addressed most of the points raised up in my first round of revision. However, the suggestion of mentioning the intrinsic properties of carbon-based materials as, in particular, graphene was to highlight the outstanding strain values achievable with nanocomposites. In the present form, such an aspect do not result to be highlighted with respect to the first version. Both the suggested references indicate that graphene can sustain relative low strain percentage with respect to nanocomposites. In particular, authors added the work by S. Zhang et al. where graphene was pressurized to reach 0.6% of tensile strain. To further emphasize the message I think you should consider as well the other work (Small 2021, 17, 2104487 https://doi.org/10.1002/smll.202104487) where even lower levels of strain were induced simply by the underlying corrugated substrate. In this way, a more quantitative evaluation of the strain achievable with carbon based materials would be provided, thus accentuating the nanocomposites features. On the contrary, the reference by B. Zhou et al. added in the same point deals with nanocomposites and it is not coherent with the issue raised; I suggest to move it before (e.g. at the end of the previous sentence).

Moreover, references numbering is not updated. Please, revise it.

Author Response

Reviews to Authors

Authors addressed most of the points raised up in my first round of revision. However, the suggestion of mentioning the intrinsic properties of carbon-based materials as, in particular, graphene was to highlight the outstanding strain values achievable with nanocomposites. In the present form, such an aspect do not result to be highlighted with respect to the first version. Both the suggested references indicate that graphene can sustain relative low strain percentage with respect to nanocomposites. In particular, authors added the work by S. Zhang et al. where graphene was pressurized to reach 0.6% of tensile strain. To further emphasize the message I think you should consider as well the other work (Small 2021, 17, 2104487 https://doi.org/10.1002/smll.202104487) where even lower levels of strain were induced simply by the underlying corrugated substrate. In this way, a more quantitative evaluation of the strain achievable with carbon based materials would be provided, thus accentuating the nanocomposites features. On the contrary, the reference by B. Zhou et al. added in the same point deals with nanocomposites and it is not coherent with the issue raised; I suggest to move it before (e.g. at the end of the previous sentence).

Moreover, references numbering is not updated. Please, revise it.

Response: We greatly appreciate this reviewer’s comments in the second round, which help us improve the manuscript a lot. We have revised our manuscript accordingly,with corresponding text edits in the manuscript highlighted in blue. References numbering has been updated.

Reviewer 2 Report

Paper is improved, but still there are major changes to be made

1. Related work section is missing, so authors are recommended to add this section by highlighting the research gaps of exisiting works, and significance of their conducted research. Why it is highly demanding?

2. Introduction section mustbe extended and rewritten by addressing the contribution

3. Abstract must present the research results and contribution in a better and clear way.

Overall paper needs major changes 

Author Response

Reviews to Authors

Paper is improved, but still there are major changes to be made.

Response: We greatly appreciate this reviewer’s comments in the second round, which help us improve the manuscript a lot. We have revised our manuscript accordingly. Below is our response to each concern of this reviewer, with corresponding text edits in the manuscript highlighted in blue.

  1. Related work section is missing, so authors are recommended to add this section by highlighting the research gaps of exisiting works, and significance of their conducted research. Why it is highly demanding?

Response: We have added some more descriptions in the “introduction” section to introduce the previous studies including the reviewer-recommended one to figure out the research gaps and further highlight the novelty of our present work. The corresponding changes have been presented as follows.

Electrospinning is a simple and efficient method of generating multifunctional nanofibers [1]. Due to their large specific surface areas, fine diameters and commendable mechanical flexibility, electrospun nanofibers have been employed to fabricate strain sensors by incorporating with conductive fillers [2-4]. However, previously reported strain sensors based on electrospun fibers were generally only capable of sensing small strains since the randomly oriented nanofibers in those sensors had low stretchability. Modifying the structure of electrospun nanofibers seemed to be a good way to improve the permanence of as-obtained electrospun sensors. For instance, the previously-reported TPU/CNTs fibrous strain sensors that were fabricated from electrospun TPU fibers with aligned wave-like structures exhibited a wide strain sensing range of 0-900% [5]. However, their full sensing range comprised three linear regions with the highest gauge factor of 20 occurring in the 600–900% strain range, which was still pretty low. Therefore, the design and development of electrospun TPU-based stain sensors with a wide strain sensing range, high linearity over this strain range, and high gauge factor still remains a huge challenge. To address the above-mentioned issues, CNTs coated TPU fibrous tubes consisting of oriented wave-like structured fibers were fabricated by a combination of electro-spinning, spraying and ultrasonic assisted deposition in the present study. The amount of CNTs deposited on the electrospun TPU fibrous tube could reach to 12 wt%, which can potentially lead to the formation of an excellent conductive net-work with a high conductivity of 0.01 S/cm. The as-prepared strain sensors exhibited a wide strain sensing range of 0-760% and importantly high linearity over the whole sensing range while maintaining high sensitivity with a GF of 57. Moreover, the strain sensors were capable of detecting a low strain (2%) and achieved fast response time whilst retaining a high level of durability. The TPU/CNTs fibrous tube-based strain sensors were found capable of accurately monitoring both large and small human body motions. Also, the strain sensors exhibited rapid response time (e.g. 45 milliseconds) combined with reliable long-term stability and durability when subjected to 60 min of water washing. The strain sensors developed in this research had the ability to detect large and subtle human motions (e.g. bending of the finger, wrist and knee, and swallowing). Consequently, this work provides an effective method for designing and manufacturing high-performance fiber-based wearable strain sensors, which offers wide strain sensing ranges and high linearity over broad working strain ranges.

References

[1] Liu, J.;  Li, T.;  Zhang, H.;  Zhao, W. W.;  Qu, L. J.;  Chen, S. J.; Wu, S. H., Electrospun strong, bioactive, and bioabsorbable silk fibroin/poly (L-lactic-acid) nanoyarns for constructing advanced nanotextile tissue scaffolds. Mater Today Bio 2022, 14.

[2] Li, T.;  Sun, M. C.; Wu, S. H., State-of-the-Art Review of Electrospun Gelatin-Based Nanofiber Dressings for Wound Healing Applications. Nanomaterials-Basel 2022, 12 (5).

[3] Zhou, Y.;  He, J.;  Wang, H.;  Qi, K.;  Nan, N.;  You, X.;  Shao, W.;  Wang, L.;  Ding, B.; Cui, S., Highly sensitive, self-powered and wearable electronic skin based on pressure-sensitive nanofiber woven fabric sensor. Sci Rep 2017, 7 (1), 12949.

[4] Lu, L.;  Wei, X.;  Zhang, Y.;  Zheng, G.;  Dai, K.;  Liu, C.; Shen, C., A flexible and self-formed sandwich structure strain sensor based on AgNW decorated electrospun fibrous mats with excellent sensing capability and good oxidation inhibition properties. Journal of Materials Chemistry C 2017, 5 (28), 7035-7042.

[5] Ren, M.;  Zhou, Y.;  Wang, Y.;  Zheng, G.;  Dai, K.;  Liu, C.; Shen, C., Highly stretchable and durable strain sensor based on carbon nanotubes decorated thermoplastic polyurethane fibrous network with aligned wave-like structure. Chemical Engineering Journal 2019, 360, 762-777.

  1. Introduction section mustbe extended and rewritten by addressing the contribution.

Response: We understand this reviewer’s concerns. The “Introduction” section has been rewritten to figure out the research gaps and further highlight the novelty of our present work. The detailed changes have been presented in the above question 1, and the corresponding changes have also been added in the main manuscript.

  1. Abstract must present the research results and contribution in a better and clear way.

Response: The “Abstract” section has been rewritten in a manner as the reviewer suggested. Some more descriptions about the research results and contribution have been added in the “Abstract” section. The corresponding changes have been presented as follows.

By combining spraying and ultrasonic assisted deposition, the amount of CNTs deposited on the electrospun TPU fibrous tube could reach to 12 wt%, which can potentially lead to the formation of an excellent conductive network with a high conductivity of 0.01 S/cm. The as-prepared strain sensors exhibited a wide strain sensing range of 0-760% and importantly high linearity over the whole sensing range while maintaining high sensitivity with a GF of 57. Moreover, the strain sensors were capable of detecting a low strain (2%) and achieved fast response time whilst retaining a high level of durability. The TPU/CNTs fibrous tube-based strain sensors were found capable of accurately monitoring both large and small human body motions. Also, the strain sensors exhibited rapid response time (e.g. 45 milli-seconds) combined with reliable long-term stability and durability when subjected to 60 min of water washing. The strain sensors developed in this research had the ability to detect large and subtle human motions (e.g. bending of the finger, wrist and knee, and swallowing).
